# COVID-19 in Pulmonary Artery Hypertension (PAH) Patients: Observations from a Large PAH Center in New York City

**DOI:** 10.3390/diagnostics11010128

**Published:** 2021-01-15

**Authors:** Roxana Sulica, Frank Cefali, Caroline Motschwiller, Rebecca Fenton, Anabela Barroso, Daniel Sterman

**Affiliations:** Department of Medicine, Pulmonary Critical Care and Sleep Division, New York University Grossman School of Medicine, New York University Langone Health, 462 1st Avenue CD 676, New York, NY 10016, USA; frank.cefali@nyulangone.org (F.C.); caroline.motschwiller@nyulangone.org (C.M.); Rebecca.fenton@nyulangone.org (R.F.); anabela.barroso@nyulangone.org (A.B.); daniel.sterman@nyulangone.org (D.S.)

**Keywords:** pulmonary arterial hypertension, COVID-19, SARS-CoV-2, clinical outcomes

## Abstract

Information on outcomes of COVID-19 in pulmonary arterial hypertension (PAH) patients is limited to a few case series and surveys. Here, we describe our experience at a large Pulmonary Hypertension Center in New York City at the height of the pandemic. We performed a retrospective chart review of eleven consecutive PAH patients who were diagnosed with SARS-CoV-2 infection. We analyzed demographics, PAH severity, risk factors for COVID-19, and COVID-19 severity and outcomes. We found in our sample that 63.6% of patients required intensive care, and there was a 45.45% overall mortality. Most patients had a known COVID-19 contact and mean duration of symptoms prior to presentation was 12 days. Only 4/11 (36%) patients presented to a center with pulmonary hypertension expertise, all of whom survived. Most patients had at least moderate pulmonary hypertension with an average REVEAL score of 7.81 despite double or triple PAH therapy. Our cases series underscores the gravity of SARS-CoV-2 infection in patients with PAH. It also suggests possible interventions to prevent unfavorable outcomes such as preserving social distancing, PAH management optimization, and early and preferential presentation to a center with specialized expertise in PAH.

## 1. Introduction

Pulmonary arterial hypertension (PAH) is a severe disease that can lead to right ventricular failure and death, with high mortality during hospitalizations both for PAH-related causes and non-cardiac conditions [1,2]. Infection with the virus SARS-CoV-2, resulting in coronavirus disease of 2019 (COVID-19), can have significant cardiopulmonary complications and mortality [3]. The outbreak of the new severe acute respiratory syndrome coronavirus-2 (SARS-CoV-2) began in December 2019 in Wuhan, China, and as a result of rapid spread, it turned into a pandemic announced by WHO on 11 March 2020. The virus is transmitted mainly through the droplet route. In most cases, it causes mild symptoms such as fever, dry cough, weakness, and muscle pain, but among elderly people, or people with impaired immunity and comorbidities, it leads to life-threatening complications in the form of pneumonitis, acute respiratory distress syndrome (ARDS), sepsis, and septic shock [4]. In the early stages of the pandemic, when both medical knowledge and testing capabilities were suboptimal, some authors suggested relatively low incidence of COVID-19 in patients with PAH and CTEPH (chronic thromboembolic pulmonary hypertension). Additionally, in published series to date, PAH patients that developed COVID-19 had seemingly favorable clinical outcomes [5,6,7,8]. Pathophysiologic mechanisms that could explain the paucity of reported cases and the surprisingly favorable outcomes of COVID-19 in patients with PAH and CTEPH have been reviewed in detail [9]. The chronic inflammatory status implicated in the pathobiology of pulmonary vascular remodeling in PAH, as well as the reduced expression ACE2 (the transmembrane receptor of the SARS-CoV-2 virus) in PAH might contribute to decreased cytokine storm and reduced viral entrance in PAH. Anticoagulation used in CTEPH could afford protective effect from the widespread prothrombotic milieu in COVID-19. In addition, specific PAH medications could have beneficial effect in SARS-CoV-2 pneumonitis and ARDS, either by hemodynamic, or anti-inflammatory mechanism [9]. Other groups have expressed concerns about the potential consequences of the optimistic message in these reports [10]. A large survey amongst center directors of Comprehensive Pulmonary Hypertension Centers showed cumulative incidence of COVID-19 in PAH/CTEPH patients similar to that in the general US population and overall worse outcomes, as well as substantial impact on the routine management of these patients [11]. A large international survey of PAH referral centers in 28 counties, identified a limited number of COVID-19 cases in patients with PAH and CTEPH, but a higher fatality rate than in the general population, and significant heterogeneity in outcomes among different countries [12].

The main aim of this work is to present clinical characteristics and outcomes of PAH patients with COVID-19 from our Pulmonary Hypertension Program at a time when New York City had been the early epicenter of the pandemic, when general medical knowledge of the disease was evolving, and Pulmonary Hypertension Center operations were particularly disrupted. We have noted a significant number of SARS-CoV-2 infected PAH patients with substantial burden of the disease and a particularly high mortality amongst patients that could not reach a center with pulmonary hypertension expertise.

## 2. Materials and Methods

We performed a retrospective chart review of the 11 consecutive patients with PAH from our program who had either confirmed or suspected COVID-19 during March–May 2020. We recorded demographic and logistical data (age, gender, BMI, sick contact or circumstances of high risk for infection), co-morbidities, clinical data on presentation (duration of symptoms, vital signs and oxygen saturation, requirement for hospital and ICU admission, need for pressors, inotropes, invasive or noninvasive ventilation, length of hospital stay), known inflammatory markers associated with COVID-19, data regarding PAH etiology, severity of PAH at the last assessment, and medical therapies at baseline (specific PAH drugs, anticoagulation, immunosuppressive agents). Finally, we assessed survival status at the end of the disease course, as well as the presence of potential cardiopulmonary sequelae at 3–6 months follow-up.

Data collection was approved by the NYU Langone IRB (i20-01694).

## 3. Results

Out of over 350 PAH/CTEPH patients followed in our program, we reviewed 10 patients with confirmed COVID-19 either by PCR at the time of the acute episode or by subsequent antibody testing after an infectious-inflammatory episode without alternative etiology. We also included one PAH patient with suspected COVID-19 based on high inflammatory markers and compatible clinical scenario despite a negative PCR test. Three of the 11 patients studied were admitted to outside institutions, leading to an incomplete data acquisition. Two of the three patients presenting at outside institutions had connective tissue disease PAH (CTD-PAH) and both underwent endotracheal intubation for progressive hypoxic respiratory failure. One patient with CTD-PAH died from refractory right ventricular failure, while the other had cardiopulmonary arrest in the setting of complicating pneumothorax. The third patient had HIV-associated PAH and died from complications of acute renal insufficiency. Complete data regarding the acute COVID-19 episode are available for the remaining eight patients, as described below.

### 3.1. Baseline Demographic and Clinical Data

Overall, there were four men and seven women, with a mean age of 59.54 years (SD = 10.53). Virtually all patients either had direct contact with a COVID-19 patient (family, co-worker, caregiver etc.), and/or resided in a ZIP code with high prevalence of COVID-19 or a Nursing Home. Out of the 11 patients, only 4 patients were able to come to the Pulmonary Hypertension Center, and only 2 patients attempted to make clinical contact to inform the program prior to presenting to the emergency room.

All patients were non-smokers with an average BMI of 30.59 kg/m^2^ (SD = 8.83). The prevalence (number of patients and percent) of co-morbidities known to be associated with severe COVID-19 is as follows: 5/11 (45%) systemic hypertension, 4/11 (36%) diabetes mellitus, 3/11 (27%) chronic kidney disease, 2/11 (18%) chronic cardiac disease (both patients with atrial fibrillation), 2/11 (18%) with other systemic disease (cirrhosis and morbid obesity with BMI of 53 kg/m^2^). Mild pre-existing parenchymal lung abnormalities were present in 5/11 (45%) of patients, with four patients with interstitial lung disease due to connective tissue disease and one patient had HIV-related obstructive lung disease.

### 3.2. Severity of PAH at Last Assessment, and Relevant Medication History

Clinical, hemodynamic, and echocardiographic data at last assessment were available in all patients. Two patients had idiopathic PAH (IPAH) and nine patients had associated PAH; 6/11 (54.5%) patients with connective tissue disease, 2/11 (18.25%) patients with HIV, and one patient with portopulmonary hypertension.

The mean duration of PAH was 4.54 years (SD = 2.56), and the last assessment was performed on the average 3.5 months (SD = 1.8) prior to the COVID-19 episode.

Table 1 describes the severity of PAH data and Table 2 shows relevant medication history on presentation.

### 3.3. COVID-19 Severity

Presenting symptoms consistent with COVID-19 were increased shortness of breath in 6/11 patients, fever/chills, and cough in 5/11, combination of shortness of breath, cough, and fevers in 6/11; one patient with portopulmonary hypertension has been transferred from the Nursing Home to the hospital with fevers and change in mental status and was diagnosed with intracerebral bleed and co-existing SARS-CoV-2 infection. Duration of symptoms prior to presentation was on average 12 days (SD = 6.59) with a median of 14 days (range 3–21 days). Out of 11 patients, 7 required ICU (intensive care unit) level of care, two were hospitalized on a regular hospital floor, and two were discharged home from the Emergency Room. Overall hospitalization rate was 81.8%, and ICU admission rate was 63.6%. Two patients underwent mechanical ventilation and two other patients were placed on high flow oxygen nasal cannula and/or 100% nonrebreather mask. All but three patients had increased requirements of oxygen supplementation but were able to maintain oxygenation above 90% with 3–6 L/min regular nasal cannula. Radiographic data were available in the nine patients admitted to the hospital, and all had increased bilateral opacities. Upon Emergency Room presentation only one patient had systemic hypotension, with two additional patients becoming hypotensive during the hospital stay. Pressors were required in one patient and inotropes in two. Average length of stay for patients admitted to the hospital was 6.9 days (SD = 6.24), median of 8 days (range 1–18 days).

Baseline respiratory data are summarized in Table 3. Clinical and laboratory data summarizing the severity of COVID-19 are depicted in Table 4. Table 5 includes hospitalization and outcome data.

Three patients received hydroxychloroquine 200 mg BID with azithromycin for seven days, while one patient received tocilizumab.

### 3.4. Clinical Outcomes

Overall, there were 5 deaths in our 11-patient cohort; with a mortality rate of 45.45%. Four out of five deaths were directly attributable to COVID-19 (mortality 36.36%). Of the COVID-related deaths, one was from progressive hypoxia/ARDS, one from mechanical complications of endotracheal intubation and mechanical ventilation (pneumothorax in ARDS), one from complications of acute kidney insufficiency, and one from decompensated right heart failure. The patient who died from right heart failure had the highest REVEAL score (11) despite triple therapy for PAH. The fifth death was in a patient with portopulmonary hypertension who had an inoperable intracerebral bleed, which was deemed at the time unrelated to the COVID-19 status.

All of the patients admitted to a center with PAH expertise survived. All of the six patients with connective tissue disease were receiving immunosuppressants for their baseline disease (mycophenolate, prednisone, or hydroxychloroquine) and only one of those patients (on hydroxychloroquine alone) developed ARDS and died. Two out of three patients with high-risk REVEAL score (equal or above 9) died, and all of the patients (*n* = 4) with low-risk REVEAL score (equal or below 6) survived. There were four patients with intermediate risk REVEAL score (7 or 8) and three of them died.

The six surviving patients have recovered without sequelae at 3- to 6-month follow-up.

## 4. Discussion

In this retrospective case series of 11 PAH patients that developed COVID-19 at the beginning of the SARS-CoV-2 pandemic in New York City, we have found a remarkably high COVID-related mortality rate of 36.36%. Causes of death were either known pathophysiological consequences of the SARS-CoV-2 viral infection (e.g., acute hypoxic respiratory failure from COVID-19 pneumonitis/ARDS or acute kidney insufficiency), or precipitation of right heart failure in a patient with reduced cardiopulmonary reserve. In addition, a large percentage of the PAH patients infected with the SARS-CoV-2 virus have developed severe COVID-19, with an overall hospitalization rate of 81.81% and requirement for ICU level of care in 63.63%. As judged by inflammatory marker abnormalities, severity of COVID-19 in this group of PAH patients was significant, although the group was heterogenous, with outliers in both directions. Despite grave outcomes for the group, heterogeneity has been noted at clinical level as well, as illustrated by two of the patients, both young women with connective tissue disease PAH, on background immunosuppressive therapy and well controlled pulmonary vascular disease on dual therapy, who had mild enough COVID-19 to be discharged home from the Emergency Room.

A small survey and other COVID-19 PAH/CTEPH series reported to date [5,6,7,8] have suggested that COVID-19 has a relatively benign course in the setting of PAH and surprisingly favorable clinical outcomes. Postulated mechanisms for the presumed low risk for severe COVID-19 included protective effects of known low ACE2 levels in PAH, resulting in decreased viral entrance in the lung cells, mitigation from the COVID-19 cytokine storm afforded by the chronic inflammatory milieu of PAH, or theoretical (and controversial) hemodynamic beneficial effects on the ventilation/perfusion matching in COVID pneumonia/ARDS of specific PAH medication [5,6,7,8,9]. A larger survey of 58 pulmonary hypertension comprehensive centers in the US showed more sobering findings with an estimated hospitalization rate of PAH/CTEPH patients with recognized COVID-19 of 30% and mortality of 12%, which are worse outcomes compared to those in general population [11]. An international survey has been undertaken in 47 PAH referral centers in 28 countries in the middle of the pandemic surge, and there were 70 cases of COVID-19 identified amongst patient with PAH and CTEPH. The observed case-fatality rate was 19%, which is a higher than in general population. Compared with the US survey, a higher percent of 70% of patients required hospitalization, although the majority were cared for on general hospital wards. Overall mortality was 20% for PAH patients and 14% for patients with CTEPH, however it was significant heterogeneity in outcomes amongst countries, with some centers reporting figures comparable to ours. [12] The significantly poor outcomes in our small series have a number of potential explanations and interpretations. At the height of the pandemic, it has been virtually impossible to ask the Emergency Medical Services to reroute the patients to the Pulmonary Hypertension Center or to transfer them once admitted in outside hospitals. The majority of our patients were not able to reach a center with specific PAH expertise, and 7/11 patients (63.6%) have been cared for in local hospitals. This is in contrast with cases reported in the international survey, where 69% of patients have been hospitalized in an expert center. [12] New York City had been the epicenter of the pandemic at the time when these data were collected, and when testing and hospital admissions were limited to patients with more severe disease [13]. This study includes only patients sick enough to seek hospital care, and we might be missing cases with milder or asymptomatic disease. General practice at the height of the pandemic in New York City favored early intubation and mechanical ventilation, and it is estimated that in New York City, the percentage of patients who received mechanical ventilation was more than 10 times higher than in China [13]. This could be due to either a more aggressive form of COVID-19 in New York, or to local practices, but irrespective of the cause, mechanical ventilation is in general poorly tolerated by PAH patients and both the sedative choice and ventilator management are particularly challenging and require special expertise that might have not been available [14]. All these factors might have contributed to higher mortality and driven the ICU admissions percentage higher in our series.

The severity of PAH and right ventricular disease at the time of the SARS-CoV-2 infection likely influenced outcomes, as all patients with low REVEAL score survived, and five out of seven patients with intermediate and high-risk REVEAL score have died. The one patient in our series with REVEAL score of 10 who survived was managed at her home institution and required ICU admission. This patient had scleroderma PAH on mycophenolate, macitentan, and riociguat for four years, and was undergoing up titration of SQ Treprostinil (dose 53 ng/kg/min on presentation) at the time of COVID-19 infection. This patient developed severe hypoxic respiratory failure and hypotension, requiring 100%FIO2/40 L/min high flow nasal cannula, low-dose norepinephrine drip, and rapid up titration of SQ Treprostinil. She survived the episode and recovered without sequelae. Of note, in hospitalized PAH patients with COVID-19 pneumonia in our cohort, the most recent REVEAL score (prior to hospital admission) and presence of baseline RV dysfunction (at the most recent PH clinic assessment) have a profound prognostic significance. This observation is likely explained by the amount of cardiopulmonary reserve at the time of COVID-19, as any added RV load (hypoxemia, hypotension, etc.) could have translated into higher mortality in patients with more severe pulmonary hypertension and RV dysfunction when compared to the other patients who had a lower REVEAL score and no RV dysfunction prior to hospital admission.

Another factor that might have resulted in poor outcomes in our series is the substantial disruptive impact that the pandemics had over the general operations of the pulmonary hypertension center, with less clinical evaluations, less diagnostic testing, and less therapeutic interventions [11,15,16,17]. The pandemic also could have contributed to the patients’ reluctance to seek medical attention, consistent with a long duration (mean of 12 days, median of 14 days) of COVID-19 symptoms prior to presentation. Furthermore, and in consensus with the large US survey data, experimental therapies for COVID-19 have not been used consistently in our series, given the lack of randomized controlled data at that stage [11].

Overall, this study is limited by the retrospective nature of the report with a small sample size and some missing data from outside centers. However, we can still learn from our experience that early presentation to a center with special PAH expertise, as well as attempts at optimization of the PAH therapy were associated with better outcomes. Every patient in our series had a known and close COVID-19 contact, which underscores the importance of strict social distancing. Severity of COVID-19 infection varied broadly in our group, but prognosis was particularly grim in patients hospitalized at non-PAH centers and those with high REVEAL scores. It is possible that background immunosuppressive therapies in connective tissue PAH attenuates to some extent the inflammatory response to COVID-19 and might result in less severe form of disease, however data to either support or refute this were not found in this study.

## 5. Conclusions

COVID-19 in PAH patients can present significant management challenges and may be associated with severe cardiopulmonary complications and high mortality. Early presentation after symptom development, PAH therapy optimization, ICU care with expertise in right heart failure management, and right ventricular protective ventilation may improve overall outcome.

## Figures and Tables

**Table 1 diagnostics-11-00128-t001:** Cause and severity of pulmonary arterial hypertension (PAH) at last assessment (*n* = 11): right heart catheterization, REVEAL score, and echocardiogram.

Patient	Age (Years)/ Gender	Type PAH	mPAP (mmHg)	RAP (mmHg)	PVR (Wood Units)	CI (L/min/m^2^)	MvO_2_ (%)	REVEAL Score	RVD	RAD
1	60 F	CTD	37	7	4.74	3.47	65	6	no	no
2	35 M	HIV	36	10	13.4	1.8	56	8	yes	yes
3	69 M	HIV	32	8	3.9	3.2	83	7	no	no
4	72 F	CTD	23	2	6.55	1.83	68	7	yes	no
5	70 M	CTD	32	12	15.13	1.42	65	11	yes	yes
6	61 F	PoPH	32	7	6.68	2.55	53	10	yes	yes
7	64 M	IPAH	60	11	10.7	2.29	65	6	no	no
8	61 F	IPAH	45	8	4.97	2.57	66	8	yes	yes
9	50 F	CTD	21	3	3.4	2.7	77	6	no	no
10	55F	CTD	20	8	1.8	3.6	74	4	no	no
11	58 F	CTD	40	17	6.42	2.83	60	10	yes	yes

F = female gender; M = male gender; CTD = connective tissue disease; HIV = human immunodeficiency virus; PoPH = portopulmonary hypertension; mPAP = mean pulmonary artery pressure; RAP = right atrial pressure; PVR = pulmonary vascular resistance; CI = cardiac index (by thermodilution); MvO2 = mixed venous oxygen saturation; RVD = right ventricular disease (dilatation and/or hypokinesis); RAD = right atrial dilatation.

**Table 2 diagnostics-11-00128-t002:** Medications.

Patient	Prostacyclin	ERA	NO Pathway	Immunosuppressant	Anticoagulation
1	Treprostinil SQ	Macitentan	Riociguat	yes	no
2		Ambrisentan	Riociguat	no	no
3		Ambrisentan	Tadalafil	no	no
4		Ambrisentan	Tadalafil	yes	no
5	Treprostinil PO	Ambrisentan	Tadalafil	no	yes
6		Macitentan		no	no
7	Treprostinil PO		Tadalafil	yes	no
8		Macitentan	Riociguat	no	yes
9	Treprostinil PO	Macitentan		yes	no
10		Macitentan	Riociguat	yes	yes
11	Treprostinil SQ	Macitentan	Riociguat	yes	no

ERA = endothelin receptor antagonist; NO = nitric oxide.

**Table 3 diagnostics-11-00128-t003:** Baseline respiratory system data.

Patient	O_2_ Supplementation (L/min)	FEV1 (%)	FVC (%)	FEV1/FVC	Radiography (CXR/CT Chest)
1	no	88	92	0.78	Normal parenchyma
2	no	84	80	0.79	Normal Parenchyma
3	2 L/min	63	65	0.64	Mild emphysema
4	2.5 L/min	65	62	0.84	NSIP
5	2–3 L/min exertion	68	69	0.78	Faint GGO, mild hilar LN
6	no	84	90	0.86	Normal parenchyma
7	no	69	67	0.81	Normal parenchyma
8	no	68	72	0.79	Normal parenchyma
9	no	64	61	0.86	NSIP
10	no	79	91	0.71	Normal parenchyma
11	no	59	56	0.80	Faint GGO

CXR = chest radiography; CT chest = computer tomography of the chest; NSIP = nonspecific interstitial pneumonitis; GGO = ground glass opacities; LN = lymphadenopathy.

**Table 4 diagnostics-11-00128-t004:** Severity of COVID-19 on presentation to the Emergency Room.

Patient	BP (S/D) (mmHg)	HR (BPM)	O_2_ Saturation (%)	D-Dimer (mcg/mL)	CRP (mg/L)	Ferritin (ng/mL)	LDH (U/L)	CXR
1	129/75	113	89	8,21	193.72	1298	446	Bilateral opacities
2	101/77	96	85	855	500	453	304	Bilateral opacities
3	NA	NA	NA	NA	NA	NA	NA	Bilateral opacities
4	NA	NA	NA	NA	NA	NA	NA	Bilateral opacities
5	NA	NA	NA	NA	NA	NA	NA	Bilateral opacities
6	122/84	60	95	3088	12.4	141	476	Bilateral opacities
7	110/66	85	89	896	116	125	401	Bilateral opacities
8	151/71	79	60	202	209.45	248	302	Bilateral opacities
9	126/75	103	91	ND	ND	ND	ND	ND
10	140/109	101	97	ND	ND	ND	ND	ND
11	80/40	100	87	691	61.1	461	273	Bilateral opacities

BP = blood pressure; S/D = systolic/diastolic; BPM = beats per minute; CRP = C-reactive protein; LDH = lactate dehydrogenase; NA = not available; ND = not done

**Table 5 diagnostics-11-00128-t005:** Hospitalization and outcome data.

Patient	PH Center	Level of Care	LOS (Days)	Pressor/ Inotrope	O_2_/Ventilatory Support	Survival Status	Cause of Death	Follow-Up (3–6 Months)
1	yes	ICU	15	yes	3 L/min (^)	alive		recovery
2	no	General floor	2	no	6 L/min (^)	alive		recovery
3	no	ICU	18	no	2 L/min (-)	dead	AKI	
4	no	ICU	1	yes	MV ETT	dead	ARDS/PTX	
5	no	ICU	8	no	MV ETT	dead	RV failure	
6	no	ICU	11	no	none	dead	ICH	
7	yes	General floor	2	no	3 L/min (^)	alive		recovery
8	no	ICU	2	no	HFNC/NRBM	dead	ARDS	
9	no	Home			none	alive		recovery
10	yes	Home			none	alive		recovery
11	yes	ICU	8	yes	HFNC	alive		recovery

LOS = length of stay; Under O_2_/ventilatory support; (-) Unchanged from baseline; (^) Increased from baseline; MV ETT = mechanical ventilation endotracheal intubation; HFNC = high flow nasal cannula; NRBM = nonrebreather mask (100% FIO_2_); AKI = acute kidney injury; PTX = pneumothorax; RV = right ventricle; ICH = intracerebral hemorrhage.

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
