# Peer review of "COVID-19 in Pulmonary Artery Hypertension (PAH) Patients: Observations from a Large PAH Center in New York City"

_diagnostics, 2021, doi:10.3390/diagnostics11010128_

Round 1
Reviewer 1 Report
The authors have undertaken and important cross sectional analysis of a vulnerable section of the pulmonary disease cohort, patients with pulmonary arterial hypertension. This article is well written and summarizes the experience of their center with COVID-19. I think this communication is valuable in improving our understanding surrounding the management of COVID 19 in PAH patients.
Major comments:
1) Table 3 is not very helpful to the reader. I would suggest that the authors discard this table and create a new "table 3" using the format used for tables 1 & 2. A patient-wise representation of the clinical data at the time of presentation would add use to this very important work. Suggested parameters include but are not limited to: oxygen parameters, use of prior supplemental oxygen, FVC, FEV1, FEV1/FVC measured prior to the presentation, REVEAL score prior to presentation, inflammatory markers at the time of presentation, admission to PAH center, floor/ICU status, vasopressor/inotrope use,length of stay, mortality and status at 3-6 months.
2) The comment about the association between the most recently assessed REVEAL score (prior to presentation) and mortality needs to be clarified. It was reported that a reveal score below 7 (n=4 patients) was protective in their case series. So are we to assume that 5/7 patients with a reveal score > 7 died ?. If this is true, can the authors please state that.
Minor comments:
Data is the plural form of datum so I would suggest replacing the verb "is" with "are" when you use the word data.
Author Response
Dear reviewer, thank you so very much for your valuable suggestions.
I have attached a corrected version of the manuscript (with track changes) incorporating all your recommendations.
- I have removed table 3 and replaced it with tables 3 (new), and 4a and 4b, pages 5-6. I truly hope that you find acceptable the way I have structured data in those 3 tables; I have included all parameters that you have recommended. REVEAL score is mentioned in table 1, which I have also expanded to include demographic and PAH etiology information.
- I have clarified the observations regarding REVEAL score and mortality, as instructed by you. Additions can be found on row 235/page 7 and row 291/page 8.
- Plural verb has been used for all instances where the word "data" appears; row/page: 96/3, 116/3, 161/4, 167/5, 278/8, 317/8.
Reviewer 2 Report
Authors presented retrospective study of patients with COVID-19 and pulmonary artery hypertension. Authors observed that about 64% of patients required intensive care, and mortality was 45%. I suggest add in Introduction some more information about SARS-CoV-2 and COVID-19, especially about epidemiology and clinical symptoms. You can cite the following paper https://sites.kowsarpub.com/jjm/articles/103744.html in which are these data. I recommend yours article for publication after this correction.
Author Response
Dear reviewer, thank you so much for your help with this manuscript and valuable suggestions. Please find attached the revised manuscript, with added COVID-19 epidemiologic and clinical information in the introductory portion, row 34/page 1, and corresponding reference 4. Thank you once again, I truly hope that you find the edits acceptable.
Round 2
Reviewer 1 Report
The edits look good.
I suggest that the title be changed to : COVID-19 pneumonia in hospitalized Pulmonary Arterial Hypertension (PAH) patients: Observations from a large PAH center in New York City.
The edits to tables look very good. With the addition of the new variables, the reader will have a clearer understanding of the severity of PAH at baseline and also the disease course.
While this is a convenience sample of hospitalized PAH patients with COVID 19 pneumonia, it is very striking to me that the most recent REVEAL score (prior to hospital admission) and presence of baseline RV dysfunction (at the most recent PH clinic assessment) have a profound prognostic significance in this cohort. This would make sense as any added RV load (hypoxemia, hypotension, etc) could have translated into mortality when compared to the other patients who had a lower REVEAL score and no RV dysfunction prior to hospital admission. The authors could highlight this in their discussion section as it is a good illustration of the prognostic effect of the REVEAL score in this patient population.
Otherwise, this article reads very well in its current form
Author Response
Dear Reviewer, thank you so much for the valuable suggestions, I have incorporated a paragraph (row 299 page 8) that stresses the importance of baseline PH severity in the overall outcome. I wholeheartedly agree, it makes sense that COVID-19 is like any other physiologic insult to the precarious cardiopulmonary reserve of the PAH patient, with potentially grave consequences.
I have no objections to changing the title to "COVID-19 Pneumonia" and "Hospitalized" patients. My only question comes from the fact that 2/11 patients (the ones with mild disease discharged from the emergency room) did not have a CXR or blood drawn for inflammatory markers. Unfortunately this has been the practice in suspected mild COVID-19 in New York City at the time of the surge. If you feel strongly about the title change, as I've mentioned, I have no objection to this effect.
Thank you once again for your review of the manuscript, respectfully, Roxana Sulica